# SPMDM: Enhancing Masked Diffusion Models through Simplifying Sampling Path

**Yichen Zhu** [1,3,*]    **Weiyu Chen** [2,*]    **James Kwok** [2]    **Zhou Zhao** [1,3,†]
[1]Zhejiang University    [2]HKUST    [3]Shanghai Artificial Intelligence Laboratory
{yc_zhu, zhaozhou}@zju.edu.cn
{wchenbx, jamesk}@cse.ust.hk

## Abstract

Autoregressive models (ARMs) show strong capabilities in many domains but face challenges with planning and complex reasoning due to their sequential generation. Masked diffusion models (MDMs) address these issues by enabling controllable, any-order, and parallel generation but encounter training difficulties as token prediction complexity varies with unmasked token positions. This work identifies two key characteristics of efficient MDM sampling paths: prioritizing tokens near unmasked ones and generating subsequence earlier in reasoning. We propose the Simple Path Masked Diffusion Model (SPMDM), which partitions sequences into fixed-length, non-overlapping subsequences and applies varying noise scales to learn token-level and cross-subsequence dependencies. Experiments on synthetic data and tasks like Countdown and Sudoku show SPMDM captures structural rules effectively, significantly outperforming existing MDMs and ARMs, with competitive results on broader reasoning benchmarks.

## 1 Introduction

Autoregressive models (ARMs) have ushered in a new era of artificial intelligence, demonstrating remarkable performance in applications such as high-quality text generation [33, 3, 43, 7], code generation [35], and chain-of-thought (CoT) reasoning for mathematical problem solving [45]. Despite the remarkable achievements and widespread real-world applications of autoregressive models, the left-to-right generation paradigm continues to suffer from several inherent limitations [21, 40, 8, 20, 46]. Notable challenges include difficulties in future planning, complex reasoning and self-correction [22, 6, 47]. These constraints have spurred researchers to explore alternative architectures for next-generation LLMs.

A compelling direction in current research focuses on the development of masked diffusion models (MDMs) [5, 26, 25, 30, 36]. Researchers have drawn inspiration from the remarkable success of diffusion models in image generation [19, 42, 34, 14] and have increasingly focused on exploring MDMs for discrete sequence generation. In contrast to AR models, which generate tokens sequentially, a unifying insight across these models is the potential of diffusion language models (DLMs) for controllable [29, 44], any-order, and parallel text generation [28, 16, 27]. Notably, DLMs exhibit promising capabilities in complex reasoning and global planning [48, 51], thereby addressing key limitations inherent in the AR approach.

However, training MDMs is inherently more challenging than training ARMs [23]. While ARMs aim to predict the next token given an unmasked prefix, MDMs are tasked with predicting a token conditioned on a set of unmasked tokens located at arbitrary positions. In other words, unlike ARMs,

---

[*]Equal contribution.
[†]Corresponding author.

39th Conference on Neural Information Processing Systems (NeurIPS 2025).

where the learning difficulty remains consistent, the denoising difficulty in MDMs varies depending on the positions of the unmasked tokens. As a result, many existing studies focus on identifying simpler sampling paths to improve sample quality. This is typically achieved by carefully selecting which token to unmask next during the sampling process [23, 32].

In this work, we begin by comparing the sampling paths, which refer to the token unmasking orders produced by different sampling strategies, and identify two key characteristics that define simpler and more efficient sampling paths: (a) From a generation-order perspective, prioritizing tokens in the local neighborhood of unmasked tokens. (b) From a logical-order perspective, prioritizing the generation of subsequences that appear earlier in the reasoning process. Motivated by these observations, we propose a novel approach called the Simple Path Masked Diffusion Model (SPMDM), which encourages these characteristics during MDM training. Specifically, we partition the input sequence into non-overlapping, fixed-length subsequences and assign different noise scales to each of them. This design encourages the model to capture token-level dependencies within each subsequence, while simultaneously learning logical dependencies across subsequences.

In our experimental evaluation, we first assess the intra- and inter-subsequence modeling capabilities of SPMDM on synthetic sequences that follow predefined structural rules. Our method significantly outperforms baseline MDMs in capturing these rules. We then focus on substantially more complex problem-solving tasks, such as Countdown and Sudoku. These tasks require extensive planning over a large number of combinations and remain challenging even for commercial large language models (e.g., GPT-4 [3]). Notably, our method significantly outperforms its autoregressive counterpart and surpasses most existing discrete diffusion language models. Finally, we evaluate SPMDM on a broader set of reasoning benchmarks, where it also demonstrates strong and competitive performance.

## 2 Background

In this section, we introduce the notation used throughout the paper and briefly review masked diffusion models (MDMs), along with a method that optimizes MDM sampling trajectories during inference via an adaptive sampling strategy. Additional related works are discussed in Appendix A.

**Notations.** We consider scalar discrete random variables with $V$ categories as one-hot column vectors in the space $\mathcal{V} = \{\mathbf{x} \in \{0,1\}^V : \sum_i x_i = 1\} \subset \Delta^V$ for the simplex $\Delta^V$. Let the $V$-th category denote a special [MASK] token, where $\mathbf{m} \in \mathcal{V}$ is its one-hot vector. We define $\mathbf{x}^{1:N}$ as a sequence of $N$ tokens, where $\mathbf{x}^\ell \in \mathcal{V}$ for all tokens $\ell \in \{1, \ldots, N\}$, and use $\mathcal{V}^N$ to denote the set of all such sequences. Throughout the work, we simplify notation and refer to the token sequence as $\mathbf{x}$ and an individual token as $\mathbf{x}^\ell$. Finally, let $\mathrm{Cat}(\cdot; p)$ be a categorical distribution with probability $p \in \Delta^V$.

### 2.1 Masked Diffusion Models

Similar to continuous diffusion models [41, 42, 19], MDMs introduce a forward process that progressively adds noise to data in the discrete domain, and learn the marginal distribution of the corresponding reverse process [36].

The forward process starts with clean data $\mathbf{x}$ and defines latent variables $\mathbf{x}_t = [\mathbf{x}_t^1, \cdots, \mathbf{x}_t^N]$ for $t \in [0, 1]$. MDLM [36] defines $q$ as a coordinate-independent masking process:

$$q_{t|0}(\mathbf{x}_t \mid \mathbf{x}_0) = \prod_{\ell=1}^{N} q_{t|0}(\mathbf{x}_t^\ell \mid \mathbf{x}_0^\ell), \; q_{t|0}(\mathbf{x}_t^\ell \mid \mathbf{x}_0^\ell) = \mathrm{Cat}(\alpha_t \mathbf{x}_0^\ell + (1 - \alpha_t)\mathbf{m}), \quad (1)$$

where $\alpha_t$ is the predefined noise schedule satisfying $\alpha_0 \approx 1, \alpha_1 \approx 0$.

The reverse process in MDMs iteratively recover values for masked tokens, starting from a mask sequence $\mathbf{x}_1 = [\mathbf{m}, \cdots, \mathbf{m}]$. Let $0 \leq s < t \leq 1$, the reverse process is given by

$$q_{s|t}(\mathbf{x}_s \mid \mathbf{x}_t, \mathbf{x}) = \prod_{\ell=1}^{N} q_{s|t}(\mathbf{x}_s^\ell \mid \mathbf{x}_t, \mathbf{x}), \; q_{s|t}(\mathbf{x}_s^\ell \mid \mathbf{x}_t, \mathbf{x}) = \begin{cases} \mathrm{Cat}(\mathbf{x}_t^\ell) & \mathbf{x}_t^\ell \neq \mathbf{m}; \\ \mathrm{Cat}\left(\frac{1-\alpha_s}{1-\alpha_t}\mathbf{m} + \frac{\alpha_s - \alpha_t}{1-\alpha_t}\mathbf{x}^\ell\right) & \mathbf{x}_t^\ell = \mathbf{m}. \end{cases}$$
$$(2)$$

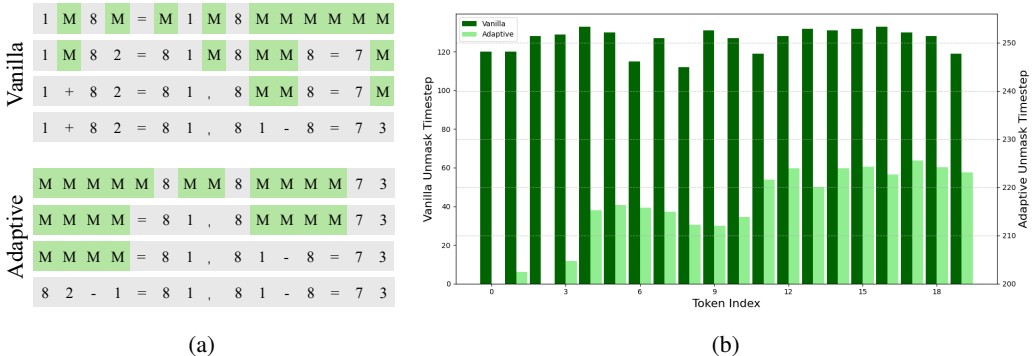

(a)                                           (b)

Figure 1: **Motivation examples.** (a) Denoising process of MDLM on the Countdown dataset using the prompt "1, 82, 8, 73,". (b) Average unmasking timestep per token index in generated chain-of-thought reasoning on the GSM8K test set.

A distribution $p_\theta(\mathbf{x} \mid \mathbf{x}_t)$ parameterized by $\theta$ is employed to approximate $q_{s|t}(\mathbf{x}_s^\ell \mid \mathbf{x}_t, \mathbf{x}) \triangleq q_{s|t}(\mathbf{x}_s^\ell \mid \mathbf{x}_t, \mathbf{x} \sim p_\theta(\mathbf{x} \mid \mathbf{x}_t))$, optimizing the following upper bound on negative log-likelihood:

$$\mathcal{L}_{\text{NELBO}} = \mathbb{E}_q \int_{t=0}^{t=1} \frac{\alpha_t'}{1 - \alpha_t} \sum_\ell \log \langle p_\theta^\ell(\mathbf{x}_t, t), \mathbf{x}^\ell \rangle \, dt. \tag{3}$$

## 2.2 Adaptive MDM Sampling Strategy

According to Equation 2, the MDM inference can be decomposed into two steps: (a) randomly selecting a set of positions to unmask and (b) assigning token values to each position via the denoising network $p_\theta$. However, unlike continuous diffusion, the reverse process in MDMs permits multiple valid sampling paths, i.e., different token unmasking orders, that are consistent with the initial distribution defined by the forward process. To identify simpler and more efficient sampling trajectories within the space of valid ones, Kim et al. [23] proposed the Adaptive MDM sampling strategy. Specifically, they designed an efficient ordering oracle function, denoted as $\mathcal{F}$, and used it to adjust the sampling process of MDMs as follows: (a) sample a set of masked tokens $\mathcal{S} = \mathcal{F}(\theta, \mathbf{x}_t) \subseteq \{\ell \mid \mathbf{x}_t^\ell = \mathbf{m}\}$, then (b) for each $i \in \mathcal{S}$, sample $\mathbf{x}_s^\ell \sim p_\theta^\ell(\mathbf{x} \mid \mathbf{x}_t)$.

## 3 Method

In this section, we begin with a motivational experiment to identify key characteristics of effective sampling paths (Section 3.1). Based on these observations, we propose Simple Path Masked Diffusion Model (SPMDM), an effective method to encourage these key characteristics during training. We provide the formulation of forward and backward processes in Section 3.2. We then discuss the network architecture in Section 3.3 and detail the training and sampling algorithms in Section 3.4.

### 3.1 Motivation

Vanilla MDM inference aims to align the intermediate distributions with the forward process, following the approach used in continuous diffusion. However, a key distinction in MDMs is that the reverse process permits multiple valid sampling paths—specifically, different token unmasking orders—all of which remain consistent with the initial distribution defined by the forward process [23].

Figure 1a illustrates two results generated by MDLM [36] trained on the Countdown dataset, employing distinct sampling strategies: vanilla and adaptive (as discussed in Section 2.2). As shown in Figure 1a, the vanilla sampling strategy does not decode semantically related tokens at close timesteps, leading to incorrect outcomes. In contrast, the adaptive sampling strategy demonstrates that groups of related tokens are decoded at closer timesteps, which contributes to achieving the correct result.

To quantitatively analyze the impact of these two sampling strategies, we conducted an experiment. Specifically, we employed a SMDM [27] fine-tuned on the GSM8K [11] dataset and compared its performance under the vanilla (original) sampling strategy and the adaptive sampling strategy. Using the GSM8K test set as input, we performed 256 sampling steps and recorded the average unmasking timestep for each token throughout the generation process. Figure 1b shows the average decoding timestep for tokens at each index. As illustrated, under the vanilla sampling strategy, the unmasking timesteps are distributed almost uniformly across all tokens. In contrast, outputs generated using the adaptive sampling strategy exhibit a distinct grouping pattern: (i) Tokens in the first half of the sequence, corresponding to the initial part of the problem (e.g., the first mathematical equation), demonstrate similar average decoding timesteps. Likewise, tokens in the second half, corresponding to the subsequent part (e.g., the second equation), also show similar average decoding timesteps among themselves. (ii) Tokens from the first half are generally decoded earlier than those in the second half, highlighting a sequential decoding pattern that aligns with the inherent structure of the problem, such as solving equations in order.

Based on the above observations, we summarize the two characteristics of simple sampling path as follows: (a) From a generation-order perspective, it prioritizes tokens in the local neighborhood of unmasked tokens ($\mathbf{x}^i \neq \mathbf{m}$). (b) From a logical-order perspective, it prioritizes the generation of subsequences that appear earlier in the reasoning process.

These characteristics are consistent with human problem-solving strategies, where we focus on each equation individually and solve them sequentially. However, the adaptive sampling strategy is applied only during inference. *Can we encourage such characteristics during training to further improve performance?*

## 3.2 Subsequence Level Forward and Reverse Process

Building upon the observation in Section 3.1, our objective is to enhance two characteristics of sampling paths during training. To achieve this, we propose to partition the input sequence $\mathbf{x}$ into multiple subsequences and apply noise of varying magnitudes across these subsequences.

Specifically, the input token sequence $\mathbf{x}$ (of total length $N$) is divided into $K$ non-overlapping subsequences, each of length $L$. Thus, $K = N/L$, assuming $N$ is an integer multiple of $L$. The $k$-th subsequence, denoted as $\mathbf{x}^k$ for $k \in \{1, \ldots, K\}$, comprises tokens from the original sequence. For convenience, we refer to the $\ell$-th token within the $k$-th subsequence as $\mathbf{x}^{k,\ell}$.

In the standard forward process of MDMs, each token $\mathbf{x}^i$ in the input sequence $\mathbf{x}$ has an equal probability of being replaced by a [MASK] token at any given noising step $t$. In contrast, our approach introduces noise with distinct magnitudes across different subsequences. The forward process for a token within a specific subsequence then becomes:

$$q_{t_k|0}(\mathbf{x}_{t_k}^{k,\ell} \mid \mathbf{x}_0^{k,\ell}) = \text{Cat}(\alpha_{t_k} \mathbf{x}_0^{k,\ell} + (1 - \alpha_{t_k})\mathbf{m}). \tag{4}$$

Here, $\mathbf{x}_0^{k,\ell}$ is the original $\ell$-th token of the $k$-th subsequence, $\mathbf{x}_{t_k}^{k,\ell}$ is its state at noising step $t_k$, and $t_k$ is the noising step specifically assigned to the entire $k$-th subsequence $\mathbf{x}^k$. $\alpha_{t_k}$ controls the noise schedule for that subsequence.

Partitioning the sequence in this manner encourages the model to learn dependencies among tokens within each subsequence, thereby encouraging characteristic (a) (in Section 3.1). Furthermore, assigning different noise scales ($t_k$) to different subsequences prompts the model to leverage information from less noisy subsequences, which strengthens its capacity to capture logical relationships across subsequences.

Correspondingly, we define a reverse process to invert the noising process defined in Equation 4. This process adapts the formulation from Equation 2 to the subsequence level:

$$q_{s_k|t_k}(\mathbf{x}_{s_k}^{k,\ell} \mid \mathbf{x}_{t_k}^{k,\ell}, \mathbf{x}^{k,\ell}) = \begin{cases} \text{Cat}(\mathbf{x}_{t_k}^{k,\ell}) & \mathbf{x}_{t_k}^{k,\ell} \neq \mathbf{m}; \\ \text{Cat}\left(\frac{1-\alpha_{t_s}}{1-\alpha_{t_k}}\mathbf{m} + \frac{\alpha_{t_s}-\alpha_{t_k}}{1-\alpha_{t_k}}\mathbf{x}^{k,\ell}\right) & \mathbf{x}_{t_k}^{k,\ell} = \mathbf{m}. \end{cases} \tag{5}$$

Here, $s_k < t_k$ are two noising steps for the $k$-th subsequence.

We employ a distribution $p_\theta(\mathbf{x}^{k,\ell} \mid \mathbf{x}_{t_k}^{k,\ell}, \mathbf{x}_{\mathbf{t}}^{-k})$ parameterized by $\theta$ to approximate $q_{s_k|t_k}(\mathbf{x}_{s_k}^{k,\ell} \mid \mathbf{x}_{t_k}^{k,\ell}, \mathbf{x}^{k,\ell})$, and the upper bound on negative log-likelihood at subsequence level can be written as

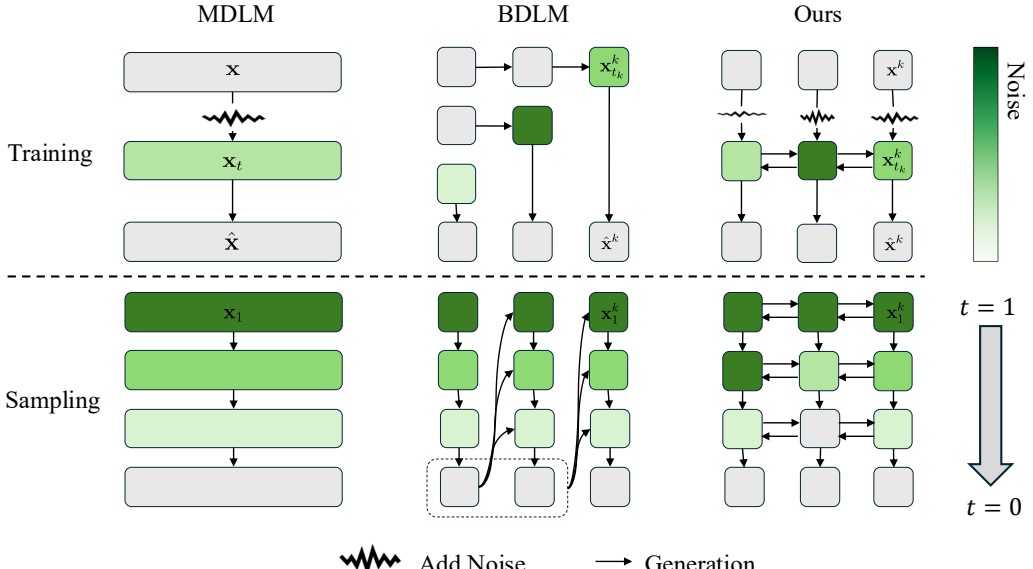

Figure 2: **Method Overview.** During training, SPMDM partitions the input sequence $\mathbf{x}$ into $K$ subsequences and introduces noise with varying magnitudes to each of them. Compared to MDLM, our method not only models token-level dependencies but also explicitly encourages the learning of inter-subsequence relationships. In contrast to BDLM, SPMDM does not impose a left-to-right generation order, enabling more flexible sampling strategies.

follows (see Appendix B for the proof):

$$\mathcal{L}_{\text{NELBO}} = \mathbb{E}_{t_k \sim [0,1], q_{t_k|0}} \left[ \frac{\alpha'_{t_k}}{1 - \alpha_{t_k}} \log p_\theta(\mathbf{x}^k \mid \mathbf{x}^k_{t_k}, \mathbf{x}^{-k}_\mathbf{t}) \right], \qquad (6)$$

where $\alpha'_t$ denotes the derivative of $\alpha_t$ with respect to $t$, and $\mathbf{x}^{-k}_\mathbf{t} = \mathbf{x}^1_{t_1}, \cdots, \mathbf{x}^{k-1}_{t_{k-1}}, \mathbf{x}^{k+1}_{t_{k+1}}, \cdots, \mathbf{x}^K_{t_K}$.

### 3.3 Network Architecture

We adopt a transformer as the base model, following existing works [25, 36]. Many discrete diffusion models do not use explicit time embeddings for the current timestep $t$ [36, 16, 27, 28]. Instead, they typically assume $t$, which indicates the noise scale applied to the input, can be implicitly inferred from the count of masked tokens.

However, the proposed method applies different noise scales to different subsequences, necessitating the introduction of time embedding layers. These layers explicitly encode the noise scale for each subsequence. In contrast to typical time embeddings in other discrete diffusion models, our design assigns $K$ distinct timestep embeddings per input sequence, one corresponding to each of the $K$ subsequences. We therefore adapt the dimensionality and structure of the timestep embedding layer for this subsequence-based approach, improving the model's capacity to capture localized noise levels within different subsequences. Note that the computational overhead from introducing these time embedding layers is minimal, as their parameter count is significantly smaller than that of other model components.

### 3.4 Training and Sampling

**Training.** We apply Equation 6 to language modeling over sequences $\mathbf{x}^{1:K}$, which are partitioned into $K$ subsequences. Specifically, we model the conditional distribution as $p_\theta(\mathbf{x}^{1:K} \mid \mathbf{x}^{1:K}_\mathbf{t}) = \prod_{k=1}^{K} p_\theta(\mathbf{x}^k \mid \mathbf{x}^k_{t_k}, \mathbf{x}^{-k}_\mathbf{t})$. Accordingly, we employ a single model to compute $p_\theta(\mathbf{x}^k \mid \mathbf{x}^k_{t_k}, \mathbf{x}^{-k}_\mathbf{t})$ for each token subsequence, and optimize the following objective:

$$\mathcal{L}_{\text{NELBO}} = \sum_{k=1}^{K} \mathbb{E}_{t_k \sim [0,1], q_{t_k|0}} \left[ \frac{\alpha'_{t_k}}{1 - \alpha_{t_k}} \log p_\theta(\mathbf{x}^k \mid \mathbf{x}^k_{t_k}, \mathbf{x}^{-k}_\mathbf{t}) \right]. \qquad (7)$$

---

**Algorithm 1** Sampling

---

1: **Input:** Network $p_\theta$, subsequence length $L$, time $[0,1]$, sampling steps $T$, oracle function $\mathcal{F}$
2: **Initialize** $\mathbf{x_1} \sim \{\mathbf{m}\}^N$, $\mathbf{t} \leftarrow \mathbf{1}$, $\Delta t \leftarrow 1/T$
3: **for** $n = 0$ to $T$ **do**
4:     $\forall k$ : Count unmasked tokens $n_k$
5:     $\forall k : t_k \leftarrow 1 - n_k/L$
6:     $\forall k : \hat{\mathbf{x}}_0^k \sim p_\theta(\cdot \mid \mathbf{x}_{t_k}^k, \mathbf{x}_{\mathbf{t}}^{-k}, t_k)$
7:     **if** using adaptive sampling strategy **then**
8:         Sample a set of masked token indices $\mathcal{S} = \mathcal{F}(\theta, \mathbf{x_t})$
9:         $\forall (i, \ell) \in \mathcal{S} : \mathbf{x}_{\mathbf{t}}^{i,\ell} = \hat{\mathbf{x}}_0^{i,\ell}$
10:     **else**
11:         $\forall k : s_k \leftarrow \max(t_k - \Delta t, 0)$
12:         $\forall k$ : For all masked tokens, with probability $\frac{s_k}{t_k}$, $\hat{\mathbf{x}}_0^{k,\ell} \leftarrow \mathbf{m}$
13:         Update $\mathbf{x_t} \leftarrow \hat{\mathbf{x}}_0$
14:     **end if**
15: **end for**
16: **Return** $\mathbf{x}_t$

---

**Sampling.** Since our forward process assigns distinct noise scales to different subsequences, we correspondingly modify the sampling procedure. Specifically, sampling starts from a fully masked sequence, denoted as $\mathbf{x}_1$. At each denoising step, for every $k$-th subsequence $\mathbf{x}^k$, we count the number of its unmasked tokens, denoted $n_k$. The corresponding timestep $t_k$ for this subsequence is then updated according to the formula $t_k = 1 - n_k/L_k$, where $L_k$ represents the length of the subsequence.

Through this timestep updating mechanism, we encourage the model to prioritize denoising tokens that are adjacent to clean tokens, fostering coherent generation. The detailed sampling process for SPMDM is depicted in Algorithm 1. It is also worth noting that our sampling method is compatible with adaptive sampling strategies. We present the corresponding ablation studies in Section 4.5 to demonstrate this compatibility.

## 4 Experiments

In this section, we first demonstrate the intra- and inter-subsequence modeling capabilities of SPMDM using toy examples in Section 4.1. We then experiment with significantly more complex problem-solving tasks in Section 4.2, which require extensive planning over a large number of combinations. Following this, we further evaluate SPMDM on a broader set of reasoning benchmarks in Section 4.3. Finally, we present an ablation study in Section 4.5. Detailed descriptions of the experimental setup are provided in the Appendix C.

### 4.1 Toy Examples

**Intra-subsequence modeling.** To assess the model's capacity to capture intra-subsequence dependencies, we construct a synthetic training dataset following a well-defined set of constraints. Each input sequence consists of 8 characters sampled from the English alphabet (i.e., a-z). Each sequence is divided into four consecutive, non-overlapping character pairs. A strict intra-pair ordering constraint is imposed: the first character in each pair must precede the second in alphabetical order. For instance, the sequence *adghklmn* is valid, as each of its four pairs, *ad*, *gh*, *kl*, and *mn*, satisfies the specified ordering constraint.

**Inter-subsequence modeling.** To evaluate the model's ability to capture inter-subsequence dependencies, we design a dataset based on a relational constraint between consecutive character pairs. Each sequence is again composed of 8 characters drawn from English alphabet, divided sequentially into four non-overlapping pairs. An inter-block ordering constraint is enforced: for any two adjacent blocks, the sum of the alphabetical positions of the characters in the preceding block must be strictly less than that of the following block. This setup encourages the model to learn and generalize the

ordering relations between blocks. For instance, the sequence *igyeporw* is valid, as it consists of four letter pairs, *ig*, *ye*, *po*, and *rw*, which satisfies the predefined ordering constraints.

**Baselines.**  In this toy example, we compare the proposed method with two representative discrete diffusion model: MDLM [36], a popular discrete diffusion model, and BDLM [4], which offers an interpolation between autoregressive models and discrete diffusion through block-wise partitioning. All methods share the same model architecture with 6M parameters, and all models are trained from scratch on the same datasets with the same number of training steps. For intra- and inter-sequence modeling, we randomly generate 50,000 samples for training and 1000 samples for testing, respectively. We use the success rate (i.e., the percentage of samples that satisfy the predefined rule) as the evaluation criterion.

**Results.**  As shown in Table 1, our method demonstrates promising performance in modeling both intra-subsequence and inter-subsequence dependencies.

For intra-subsequence dependencies, the proposed method achieves 26.6% higher success rates than MDLM and 3.4% higher than BDLM, clearly showcasing the ability of SPMDM to effectively capture intra-subsequence relationships.

Table 1: Success rates of different methods on intra-subsequence and inter-subsequence modeling.

| Method | Intra-subseq | Inter-subseq |
|---|---|---|
| MDLM | 69.2 | 58.3 |
| BDLM | 92.4 | **87.1** |
| Ours | **95.8** | 80.6 |

In terms of inter-subsequence modeling, our method also exhibits strong capabilities, outperforming MDLM by 22.3%. By assigning different noise scales to different subsequences, the method encourages the model to prioritize extracting information from subsequences with lower noise levels, thereby enhancing its ability to model inter-subsequence dependencies.

Although our method performs slightly worse than BDLM in inter-subsequence modeling, this is largely due to BDLM's use of a semi-autoregressive sampling strategy during inference. Specifically, BDLM decodes sequentially from the left block to the right block, which aligns perfectly with the requirements of this specialized task. However, as will be shown in real-world experiments, BDLM does not perform well overall.

## 4.2  Problem-solving Tasks

**Dataset.**  Countdown [1] is a mathematical reasoning task and a generalization of the classic "24 Game," which remains challenging even for advanced models such as GPT-4 [3]. The objective is to combine a given set of numbers using basic arithmetic operations $(+, -, \times, \div)$ to exactly match a specified target number. We consider three subtasks of increasing difficulty by varying the number of input digits in $3, 4, 5$. Sudoku [2] is a classic logic-based number placement puzzle, widely recognized for its stringent intellectual requirements. The objective is to fill a $9 \times 9$ grid with digits such that each row, each column, and each of the nine $3 \times 3$ subgrids contains all numbers from 1 to 9 exactly once. In our setup, the digit 0 is used to indicate vacant positions to be filled by the model. Each $9 \times 9$ grid is linearized into a sequence of 81 digits, which serves as the model input. During tokenization, each digit is treated as an individual token.

**Baselines.**  In addition to the previously introduced MDLM [36] and BDLM [4] baselines, we also include autoregressive models based on the GPT-2 architecture [33], with parameter sizes ranging from 6M to 85M, as well as pretrained large language models from the LLaMA family [43], with parameter sizes ranging from 7B to 13B. Furthermore, we compare our approach against several other existing discrete diffusion models: D3PM [5], SEDD [25], and MGDM [48]. D3PM serves as the canonical framework for discrete diffusion modeling. SEDD adapts score-based techniques from continuous diffusion into the discrete domain. MGDM exhibits strong performance on reasoning and planning tasks. Except for LLaMA, all other baselines are trained from scratch on the dataset.

**Results on Countdown.**  As shown in Table 2, our method demonstrates superior performance on the Countdown task. Notably, the performance gap widens as the number of input digits increases. We attribute this to the fact that countdown can be framed as a brute-force search problem, where a larger

Table 2: Test Accuracy on Countdown (CD).

| Method | Param | CD 3 | CD 4 | CD 5 |
|--------|-------|------|------|------|
| GPT-2 | 6M | 94.1 | 31.9 | 4.3 |
| | 85M | 95.9 | 45.8 | 5.1 |
| LLaMA | 7B | 95.7 | 44.1 | 6.7 |
| | 13B | 96.5 | 51.1 | 7.4 |
| D3PM | 85M | 99.4 | 83.1 | 27.6 |
| SeDD | 85M | 99.4 | 83.7 | 39.9 |
| MDLM | 85M | 99.5 | 85.8 | 41.2 |
| MGDM | 85M | 99.5 | 91.5 | **46.6** |
| BDLM | 85M | 98.7 | 85.5 | 40.8 |
| **Ours** ($L = 8$) | 6M | 98.3 | 53.2 | 27.0 |
| **Ours** ($L = 8$) | 85M | **99.6** | **92.7** | 46.1 |

Table 3: Test Accuracy on Sudoku.

| Method | Param | Acc (↑) |
|--------|-------|---------|
| GPT-2 | 6M | 13.1 |
| | 85M | 22.4 |
| LLaMA | 7B | 28.7 |
| | 13B | 41.2 |
| D3PM | 85M | 89.8 |
| SeDD | 85M | 90.2 |
| MDLM | 85M | 92.3 |
| MGDM | 85M | 99.9 |
| BDLM | 85M | 92.1 |
| **Ours** ($L = 9$) | 6M | 99.9 |
| **Ours** ($L = 9$) | 85M | **100.0** |

Table 4: Performance on language understanding and reasoning benchmarks. For GSM8K, we finetune the models; all other tasks are evaluated in a zero-shot setting.

| Model | Param | CommonSense Reasoning | | | | Math |
|-------|-------|-------|------|------|-------|-------|
| | | HSwag | SIQA | PIQA | Wino. | GSM8K |
| LLaMA | 7B | 74.9 | 43.2 | 63.3 | 67.1 | 58.6 |
| Plaid | 1.3B | 39.3 | 32.3 | 54.5 | 51.3 | 32.6 |
| GPT-2 | 127M | 29.9 | 35.7 | **62.1** | 48.5 | 44.8 |
| SEDD | 170M | 30.2 | 34.4 | 55.6 | 50.1 | 45.3 |
| MDLM | 127M | 31.5 | 35.0 | 54.2 | 50.4 | 46.1 |
| DiffuGPT | 127M | 33.4 | 37.0 | 57.7 | 50.8 | 50.2 |
| **Ours** ($L = 8$) | 127M | **33.8** | **37.2** | 56.9 | **51.2** | **51.3** |
| GPT-2 | 355M | **38.8** | 37.7 | **67.4** | 50.7 | 50.7 |
| SEDD | 424M | 31.5 | 35.4 | 56.1 | 49.0 | 53.5 |
| MDLM | 355M | 32.7 | 37.4 | 55.1 | 51.5 | 54.1 |
| DiffuGPT | 355M | 37.2 | 39.9 | 59.6 | 52.6 | 52.6 |
| **Ours** ($L = 8$) | 355M | 36.3 | **41.7** | 58.4 | **53.2** | **56.4** |

number of input digits leads to a combinatorially larger search space and necessitates longer-range reasoning during generation. Under such conditions, the performance of autoregressive models and semi-autoregressive baselines such as BDLM degrades due to their limited horizon. In contrast, our approach exhibits robust performance when handling long reasoning chains, benefiting from its broader context window and a stronger ability to model logical dependencies across subsequences.

**Results on Sudoku.** Unlike Countdown, the Sudoku task requires a balance between global structural consistency and fine-grained intra-subsequence constraints. We report the results on the Sudoku in Table 3. Due to the superior capability of SPMDM in capturing local patterns within subsequences, it is able to effectively learn row-level constraints. At the same time, its strength in modeling logical dependencies across subsequences enables it to learn subgrid and column-level constraints as well. In contrast, BDLM that follow a left-to-right generation paradigm tend to struggle with capturing subgrid and column-level rules effectively.

## 4.3 Reasoning Tasks

**Benchmarks.** We evaluate our model on a suite of challenging benchmarks spanning language understanding and reasoning. For common sense reasoning, we include four multiple-choice datasets: HellaSwag(HSwag) [50], SocialIQA (SIQA) [38], PhysicalIQA (PIQA) [9], and Winogrande (Wino.) [37], with accuracy as the evaluation metric. Additionally, we test on GSM8K [11], a benchmark of grade school math word problems, to evaluate the model's mathematical reasoning

capabilities. We follow Ye et al. [49] in finetuning setting using the augmented symbolic data to test the CoT [45] math reasoning abilities of diffusion models.

**Baselines.** In addition to the baselines introduced in earlier experiments, we incorporate several representative models for comparison in this section: Plaid [17], a state-of-the-art continuous diffusion-based language model; and DiffuGPT [16], a discrete diffusion model for language generation adapted from GPT-2.

**Result.** As shown in Table 4, under the same parameter budget, our SPMDMs achieve state-of-the-art performance across a variety of reasoning task benchmarks. The slightly worse performance on PIQA compared to GPT-2 may be due to the specific physical knowledge required for the task, which our models may lack. This limitation stems from fine-tuning on MDLMs trained on only 30B tokens of OpenWebText, which may be insufficient to acquire broad general knowledge. In tasks that require more extensive global reasoning, such as GSM8K, SPMDMs consistently exhibit better performance compared to AR models that rely solely on left-to-right modeling capabilities.

## 4.4 Computational Overhead

We evaluated the computational efficiency of SPMDM through speed benchmarks on the Countdown 4 dataset, utilizing a single RTX 4090 GPU. Specifically, we measured: (1) the training and evaluation token throughput of both MDLM and SPMDM with a batch size of 32, and (2) the time required for a 32-step sampling process with a batch size of 1. The results are summarized in Table 5. Note that the official MDLM implementation does not remove the timestep embedder but instead provides it with an all-zero input. Consequently, reintroducing the timestep embedding in SPMDM incurs no additional computational overhead compared to MDLM. As shown in Table 5, these results confirm that SPMDM matches the computational efficiency of baseline approaches.

Table 5: Comparison of computational efficiency between SPMDM and MDLM.

|  | MDLM | SPMDM |
|---|---|---|
| Training (token/ms) | 93.7 | 93.1 |
| Evaluation (token/ms) | 266.8 | 265.1 |
| Sampling (s) | 0.55 | 0.57 |

## 4.5 Ablation Studies

**Effect of subsequence length.** In the experiments of Section 4.2, we demonstrated the problem-solving capability of SPMDM under a fixed subsequence length. We now shift our focus to analyzing how different subsequence lengths affect model performance. Notably, when $L = N$, our model degenerates into MDLM. Specifically, SPMDM reduces to MDLM when $L = 48$ for Countdown and $L = 162$ for Sudoku. The experimental results are shown in Figure 3a. Overall, SPMDM consistently outperforms MDLM across most subsequence lengths. However, when $L = 4$, we observe a performance drop, which we attribute to the subsequences being too short to capture sufficient semantic or logical information, thus increasing the difficulty of modeling inter-subsequence dependencies. Conversely, when $L = 16$, the subsequences become too long and are likely to contain most of the solution within a single subsequence, thereby diminishing the need for modeling logical dependencies across subsequences.

**Effect of sampling strategy.** We also conduct ablation studies on different sampling strategies. The evaluated strategies include: 1) DDPM sampling, the standard method introduced in MDLM, where the same timestep $t$ is applied to the entire sequence $\mathbf{x}$ at each denoising step; 2) Our proposed sampling strategy (i.e. lines 10 to 13 in Algorithm 1), where each subsequence is assigned an individual noise scale and its timestep is dynamically updated during sampling; 3) A hybrid strategy that combines our sampling method with adaptive sampling (i.e. lines 7 to 9 in Algorithm 1). The corresponding results are presented in Figure 3b. Clearly, our sampling approach—assigning noise scales on a per-subsequence basis—leads to more effective denoising. Furthermore, incorporating

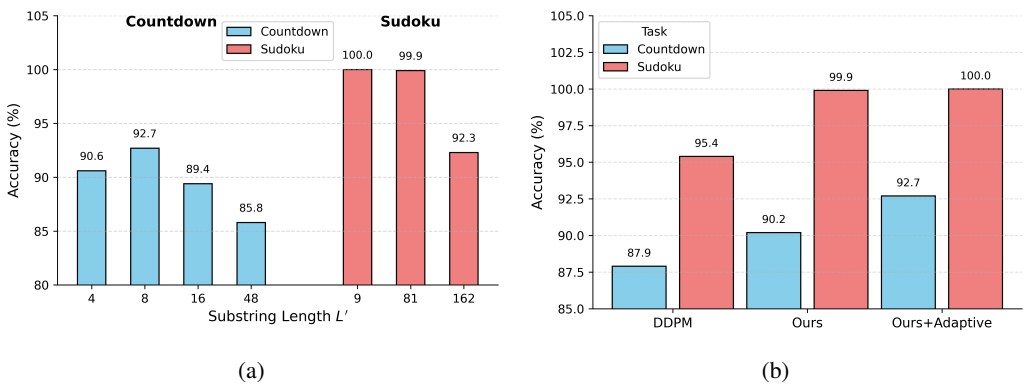

| (a) | (b) |
|---|---|

Figure 3: Ablation studies on subsenquence length $L$ and sampling strategy using SPMDM trained on countdown and sudoku datsets. (a) Accuracy on countdown and sudoku vs. subsenquence length. (b) Accuracy on countdown and sudoku vs. sampling strategy.

adaptive sampling further enhances generation quality, indicating the complementary strengths of both strategies.

## 5   Conclusion

In conclusion, we identify two key characteristics that define simpler and more efficient sampling paths during MDM sampling process. Motivated by this two characteristics, we propose a novel MDM framework, termed SPMDM, whose core idea is to encourage the characteristics of simple sampling paths during training, thereby enhancing the model's ability to discover and follow simpler trajectories during the sampling process. Extensive quantitative experiments across a variety of tasks demonstrate the effectiveness and superiority of our proposed method.

## Acknowledgment

This work was supported by National Key R&D Program of China (No.2022ZD0162000) and National Natural Science Foundation of China (No.62222211, No.U24A20326). This work was also supported in part by the Research Grants Council of the Hong Kong Special Administrative Region (Grants 16202523 and HKU C7004-22G).

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

## A    Related Works

**Autoregressive Models.**    The autoregressive modeling paradigm, where the prediction of a token only depends on the preceding context, is widely adopted in modeling language. Autoregressive models have catalyzed significant advances in artificial intelligence, achieving state-of-the-art results across a range of tasks, including fluent text synthesis [33, 3, 43, 7], program generation [35], and chain-of-thought reasoning in mathematical domains [45]. However, despite their transformative impact and broad deployment in real-world systems, ARMs are inherently constrained by their sequential, left-to-right generation strategy [21, 40, 8, 20]. This unidirectional nature presents persistent challenges in scenarios requiring foresight, multi-step reasoning, and iterative self-correction [22, 6, 47].

**Diffusion Language Models.**    Continuous diffusion models have demonstrated remarkable performance and controllability in image generation tasks [19, 42, 12, 18]. Building on these successful practices, several works have extended continuous diffusion models to text generation [24, 15, 13, 17]. Among them, Plaid [17] is a notable approach that maps discrete text into a continuous embedding space and constructs a continuous diffusion framework in that space. Given the inherently discrete nature of language, Austin et al. [5] proposed D3PM, a diffusion framework tailored to discrete data domains. They incorporate an absorbing `[MASK]` state as noise, laying the foundation for discrete diffusion models. This framework has been further developed by [25], [30], [36], and [39]. Among these, the MDLM [36] framework is one of the most widely adopted, offering a simple and efficient training objective. More recently, BDLM [4] combines ARMs with MDMs through interpolation, integrating the left-to-right generation paradigm of ARMs into MDMs and proposing a novel diffusion modeling framework. Furthermore, Ye et al. [48] have shown that MDMs significantly outperform ARMs in complex reasoning and global planning tasks.

## B    Simple Path Mask Diffusion Model

Recall that, under the SPMDM framework, the input token sequence $\mathbf{x}$ (of total length $N$) is divided into $K$ non-overlapping subsequences, each of length $L$. Thus, $K = N/L$, assuming $N$ is an integer multiple of $L$. The $k$-th subsequence, denoted as $\mathbf{x}^k$ for $k \in \{1, \ldots, K\}$, comprises tokens from the original sequence. For convenience, we refer to the $\ell$-th token within the $k$-th subsequence as $\mathbf{x}^{k,\ell}$.

### B.1    Forward Process

The forward noise process applied independently for each token is defined as follows:

$$q_{\mathbf{t}|0}(\mathbf{x_t} \mid \mathbf{x}_0) = \prod_{k}^{K} \prod_{\ell}^{L} q_{t_k|0}(\mathbf{x}_{t_k}^{k,\ell} | \mathbf{x}_0^{k,\ell}), \ q_{t_k|0}(\mathbf{x}_{t_k}^{k,\ell} \mid \mathbf{x}_0^{k,\ell}) = \mathrm{Cat}(\alpha_{t_k}\mathbf{x}_0^{k,\ell} + (1 - \alpha_{t_k})\mathbf{m}), \quad (8)$$

where $\mathbf{t} = t_1, \cdots, t_K$, and $t_k$ denotes the noising step applied to $\mathbf{x}^k$.

## B.2 Reverse Process

Under the framework of MDLM [36], the reverse process iteratively recover values for masked tokens, starting from a mask sequence $\mathbf{x}_1 = [\mathbf{m}, \cdots, \mathbf{m}]$. Let $0 \leq s_k < t_k \leq 1$, the reverse process is given by:

$$q_{\mathbf{s}|\mathbf{t}}(\mathbf{x_t} \mid \mathbf{x}_0) = \prod_k^K \prod_\ell^L q_{s_k|t_k}(\mathbf{x}_{s_k}^{k,\ell} \mid \mathbf{x}_{t_k}^{k,\ell}, \mathbf{x}^{k,\ell}),$$

$$q_{s_k|t_k}(\mathbf{x}_{s_k}^{k,\ell} \mid \mathbf{x}_{t_k}^{k,\ell}, \mathbf{x}^{k,\ell}) = \begin{cases} \mathrm{Cat}(\mathbf{x}_{t_k}^{k,\ell}) & \mathbf{x}_{t_k}^{k,\ell} \neq \mathbf{m}; \\ \mathrm{Cat}\left(\frac{1-\alpha_{t_s}}{1-\alpha_{t_k}}\mathbf{m} + \frac{\alpha_{t_s}-\alpha_{t_k}}{1-\alpha_{t_k}}\mathbf{x}^{k,\ell}\right) & \mathbf{x}_{t_k}^{k,\ell} = \mathbf{m}. \end{cases} \tag{9}$$

## B.3 Simple Path Mask Diffusion NELBO

We provide the negative evidence lower bound (NELBO) for the simple path masked diffusion parameterization. We firstly perform diffusion in each block over $T$ discretization steps. Let $\mathrm{D_{KL}}[\cdot]$ denote the Kullback-Leibler divergence, $t_k, s_k$ be shorthand for $t_k(i) = i/T$ and $s_k(i) = t(i-1)/T$, $\forall i \in [1, T]$. We derive the NELBO as follows:

$$-\log p_\theta(\mathbf{x}) = -\sum_{k=1}^{K} \log \mathbb{E}_q \left[ \frac{p_\theta(\mathbf{x}_{t_k(1):t_k(T)}^k \mid \mathbf{x_t}^{-k})}{q(\mathbf{x}_{t_k(1):t_k(T)}^k \mid \mathbf{x}^k)} \right]$$

$$= -\sum_{k=1}^{K} \log \mathbb{E}_q \left[ \frac{p_\theta(\mathbf{x}_{t_k(T)}^k \mid \mathbf{x_t}^{-k}) \prod_{i=1}^{T} p_\theta(\mathbf{x}_{s_k(i)}^k \mid \mathbf{x}_{t_k(i)}^k, \mathbf{x_t}^{-k})}{\prod_{i=1}^{T} q(\mathbf{x}_{s_k(i)}^k \mid \mathbf{x}_{t_K(i)}^k)} \right]$$

$$\leq \sum_{k=1}^{K} \left[ \underbrace{-\mathbb{E}_q \log p_\theta(\mathbf{x}^k \mid \mathbf{x}_{t_k=\frac{1}{T}}^k, \mathbf{x_t}^{-k})}_{\mathcal{L}_{\mathrm{recons}}} \right. \tag{10}$$

$$\left. + \underbrace{\mathbb{E}_{t_k \in \{\frac{2}{T}, \dots, \frac{T-1}{T}, 1\}} \mathbb{E}_q \, T \, \mathrm{D_{KL}}\left(q(\mathbf{x}_{s_k}^k \mid \mathbf{x}_{t_k}^k, \mathbf{x}^k) \,\|\, p_\theta(\mathbf{x}_{s_k}^k \mid \mathbf{x}_{t_k}^k, \mathbf{x_t}^{-k})\right)}_{\mathcal{L}_{\mathrm{diffusion}}} \right.$$

$$\left. + \underbrace{\mathrm{D_{KL}}\left(q(\mathbf{x}_{t_k=1}^k \mid \mathbf{x}^k) \,\|\, p_\theta(\mathbf{x}_{t_k=1}^k)\right)}_{\mathcal{L}_{\mathrm{prior}}} \right]$$

## B.4 Simplified NELBO

We adopt the SUBS parameterization proposed by Sahoo et al. [36]. Specifically, we impose the following constraints on the design of $p_\theta$ by exploiting the fact that, at any timestep $t$, each token $\mathbf{x}_t^\ell$ can only reside in one of two states: the original token $\mathbf{x}^\ell$ or the mask token $\mathbf{m}$, i.e., $\mathbf{x}_t^\ell \in \mathbf{x}^\ell, \mathbf{m}$ for all $\ell \in 1, \dots, L$:

1. **Zero Masking Probability.** Since the clean target sequence $\mathbf{x}$ does not contain any mask tokens, we enforce $p_\theta(\mathbf{x}^\ell = \mathbf{m} \mid \mathbf{x}_t^\ell) = 0$, ensuring that the model never predicts a mask token during denoising.

2. **Carry-Over Unmasking.** Once a token is unmasked in the reverse process, it is never remasked. Accordingly, we simplify the denoising model by enforcing $p_\theta(\mathbf{x}_s^\ell = \mathbf{x}_t^\ell \mid \mathbf{x}_t^\ell \neq \mathbf{m}) = 1$, meaning that any token already unmasked remains unchanged in subsequent steps.

Table 6: Dataset Deatails. Intra and Inter refer to toy datasets designed for intra- and inter-subsequence modeling, respectively. CD is an abbreviation for Countdown..

| | Intra | Inter | CD3 | CD4 | CD5 | Sudoku |
|---|---|---|---|---|---|---|
| Train Entries | 50k | 50k | 500k | 500k | 500k | 100k |
| Test Entries | 1k | 1k | 1k | 1k | 1k | 1k |
| Avg Input Token | 8 | 8 | 11 | 13 | 16 | 81 |
| Avg Output Token | - | - | 12 | 25 | 35 | 81 |
| Max Input Token | 8 | 8 | 16 | 15 | 18 | 81 |
| Max Output Token | - | - | 22 | 35 | 52 | 81 |

As a result, we will only approximate the posterior $p_\theta(\mathbf{x}_s^\ell = \mathbf{x}^\ell \mid \mathbf{x}_t^\ell = \mathbf{m})$. The diffusion loss term becomes:

$$
\begin{aligned}
\mathcal{L}_{\text{diffusion}} &= \sum_{k=1}^{K} \mathbb{E}_{t_k} \mathbb{E}_q T \left[ \mathrm{D}_{\mathrm{KL}} \left( q(\mathbf{x}_{s_k}^k \mid \mathbf{x}_{t_k}^k, \mathbf{x}^k) \,\|\, p_\theta(\mathbf{x}_{s_k}^k \mid \mathbf{x}_{t_k}^k, \mathbf{x}_{\mathbf{t}}^{-k}) \right) \right] \\
&= \sum_{k=1}^{K} \mathbb{E}_{t_k} \mathbb{E}_q T \left[ \sum_{\ell=1}^{L} \mathrm{D}_{\mathrm{KL}} \left( q(\mathbf{x}_{s_k}^{k,\ell} \mid \mathbf{x}_{t_k}^{k,\ell}, \mathbf{x}^{k,\ell}) \,\|\, p_\theta(\mathbf{x}_{s_k}^{k,\ell} \mid \mathbf{x}_{t_k}^{k,\ell}, \mathbf{x}_{\mathbf{t}}^{-k}) \right) \right] \\
&= \sum_{k=1}^{K} \mathbb{E}_{t_k} \mathbb{E}_q T \left[ \sum_{\ell=1}^{L} \frac{\alpha_{t_k} - \alpha_{s_k}}{1 - \alpha_{t_k}} \log p_\theta(\mathbf{x}^{k,\ell} \mid \mathbf{x}_{t_k}^{k,\ell}, \mathbf{x}_{\mathbf{t}}^{-k}) \right] \\
&= \sum_{k=1}^{K} \mathbb{E}_{t_k} \mathbb{E}_q T \left[ \frac{\alpha_t - \alpha_s}{1 - \alpha_t} \log p_\theta(\mathbf{x}^k \mid \mathbf{x}_{t_k}^k, \mathbf{x}_{\mathbf{t}}^{-k}) \right]
\end{aligned}
\tag{11}
$$

Previous works have shown empirically and mathematically that increasing the number of steps $T$ yields a tighter approximation to the ELBO [10]. Following a similar argument, we form an continuous time extension by taking $T \to \infty$, which yields the following diffusion loss term:

$$
\mathcal{L}_{\text{diffusion}} = \sum_{k=1}^{K} \mathbb{E}_{t_k \sim [0,1]} \mathbb{E}_q \left[ \frac{\alpha'_{t_k}}{1 - \alpha_{t_k}} \log p_\theta(\mathbf{x}^k \mid \mathbf{x}_{t_k}^k, \mathbf{x}_{\mathbf{t}}^{-k}) \right]
\tag{12}
$$

For the continuous time case, we have $\mathbf{x}_{t_k = \frac{1}{T}}^k \sim \lim_{T \to \infty} \mathrm{Cat}\left(\mathbf{x}_{t_k = \frac{1}{T}}^k\right) = \mathrm{Cat}(\mathbf{x}^k)$. Then, the reconstruction loss term becomes:

$$
\mathcal{L}_{\text{recons}} = -\mathbb{E}_q \log p_\theta(\mathbf{x}^k \mid \mathbf{x}_{t_k = \frac{1}{T}}^k, \mathbf{x}_{\mathbf{t}}^{-k}) = -\log p_\theta(\mathbf{x}^k \mid \mathbf{x}_{t_k = \frac{1}{T}}^k = \mathbf{x}^k, \mathbf{x}_{\mathbf{t}}^{-k}) = 0
\tag{13}
$$

The prior loss also reduces to 0 because $\alpha_{t=1} = 0$, which ensures $q(\mathbf{x}_{t_k=1}^k \mid \mathbf{x}^k) = \mathrm{Cat}(\mathbf{m})$ and $p_\theta(\mathbf{x}_{t_k=1}^k) = \mathrm{Cat}(\mathbf{m})$.

Finally, we obtain a simple objective as follows:

$$
\mathcal{L}_{\text{NELBO}} = \sum_{k=1}^{K} \mathbb{E}_{t_k \sim [0,1]} \mathbb{E}_q \left[ \frac{\alpha'_t}{1 - \alpha_t} \log p_\theta(\mathbf{x}^k \mid \mathbf{x}_{t_k}^k, \mathbf{x}_{\mathbf{t}}^{-k}) \right]
\tag{14}
$$

## C  Experimental Details

### C.1  Dataset Details

We present the detailed specifications of the toy datasets and problem-solving task datasets in Table 6.

### C.2  Implementation Details

**Toy examples.**  We conduct all toy example experiments using four RTX 4090 GPUs. MDLM [36], BDLM [4], and SPMDM are all implemented using a tiny model with 6M parameters. For BDLM, the block size is set to 2, and for SPMDM, the subsequence length is also set to 2. We use a learning rate of $1 \times 10^{-3}$ and a batch size of 1024. All models are trained for 10 epochs on the training set. Additionally, the number of sampling steps is fixed to 32 for all models.

**Problem-solving tasks.** We conduct all experiments related to problem-solving tasks using eight RTX 4090 GPUs. Both ARMs and MDMs are implemented based on the GPT-2 architecture. Across all datasets, we use a learning rate of $1 \times 10^{-3}$ for the 6M-parameter tiny models and $3 \times 10^{-4}$ for the 85M-parameter models. The batch size is set to 512. For the countdown task, we train for 150 epochs, and for the sudoku task, we train for 100 epochs. Specifically, for the Countdown task, we set the block size of BDLM to 4, and for the Sudoku task, we set it to 9. During sampling, we fix the number of denoising steps to 32 for all MDMs across all tasks. For LLaMA, we follow the results reported in the work of Ye et al. [48]; detailed fine-tuning settings can be found in Appendix C of [48].

**Reasoning tasks.** For GPT-2, SEDD, and DiffuGPT, we borrow the results reported by Gong et al. [16]. Following their experimental setup, we also use the advanced FineWeb2 corpus [31], which is derived from Common Crawl, as the training dataset for both MDLM and SPMDM. Training and sampling are conducted on eight A100 GPUs with 40GB of memory. For models with 127M and 355M parameters, we use a learning rate of $3 \times 10^{-4}$ with a cosine scheduler. The batch size is set to 512, and training is performed for a total of 400K iterations. During inference, the number of sampling steps is fixed to 256.

### C.3 Evaluation Details

For the intra- and inter-subsequence modeling tasks, we perform unconditional generation using MDMs, generating 1,000 samples per task and evaluating performance by counting the number of samples that satisfy the predefined structural constraints. For the countdown and sudoku tasks, we conduct conditional generation using the questions from the test set as prompts, and measure performance by the number of correctly solved instances. For common sense reasoning tasks—HellaSwag [50], Winogrande [37], SIQA [38], and PIQA [9]—we use answer accuracy as the evaluation metric. For GSM8K [11], we take the questions from the test set as prompts for conditional generation and report the accuracy of the final predicted answers as the evaluation metric.

## D Limitation

SPMDM framework is based on factorization assumptions. In a state space of length $N$, the transition matrix [10] contains an exponential number of possible states, making it computationally expensive to reverse. To alleviate this issue, existing works [10, 25, 36, 39] assume independence between dimensions, treating each dimension as an independent one-dimensional diffusion process with the same transition rate matrix. Admittedly, in language modeling, tokens are not entirely independent, there exist complex dependencies between them. However, without this independence assumption, the computational cost of training would become astronomical, and the modeling complexity would increase significantly. Despite this simplification, extensive prior work [10, 25, 36, 39] and our own experiments demonstrate that under this assumption, the model achieves strong performance with practically acceptable results for real-world applications. At the same time, this presents an interesting research direction—exploring ways to explicitly model the conditional dependencies between tokens. By leveraging these dependencies to strategically plan the denoising process of DDMs, we can potentially unlock significant improvements in the model's generative capabilities.

## E Impact

**Ethical impacts.** This study does not pose any ethical concerns. It does not involve subjective assessments or the use of private data, as all experiments are conducted on publicly available datasets.

**Expected societal implications.** The primary potential societal impact of SPMDM lies in its possible misuse, particularly in generating false or misleading information, which could contribute to misinformation, privacy violations, and other harmful consequences. To mitigate these risks, it is essential to establish robust ethical guidelines and implement continuous monitoring to ensure the responsible and ethical deployment of such generative models.

