# OpenReview forum: "SPMDM: Enhancing Masked Diffusion Models through Simplifying Sampling Path"
_NeurIPS.cc/2025/Conference — NeurIPS 2025 poster_

### Official Review · Reviewer_E7mR · 2025-06-01

**Clarity:** 3
**Significance:** 3
**Originality:** 3
**Rating:** 5
**Confidence:** 4

**Summary:**

This paper introduces the Simple Path Mask Diffusion Model (SPMDM), a method designed to enhance the sampling efficiency and capability of Masked Diffusion Models (MDMs) for discrete sequence generation tasks. Unlike traditional autoregressive models, MDMs support parallel, non-sequential generation, but training can be (quite) challenging due to varying prediction difficulties depending on the positions of unmasked tokens. The authors observe two key properties that characterize efficient sampling paths: (1) tokens near already unmasked positions are easier and should be prioritized, and (2) subsequences appearing earlier in logical reasoning should be decoded first. Leveraging these insights, SPMDM partitions sequences into fixed-length, non-overlapping subsequences, applying varying noise scales to explicitly model both token-level dependencies within subsequences and logical dependencies across subsequences.

Mathematically, SPMDM modifies the standard forward and reverse diffusion processes by assigning distinct noising steps to each subsequence, introducing subsequence-specific noise levels. The forward process masks tokens differently across subsequences, and the reverse process denoises them accordingly. The training objective thus explicitly captures subsequence-level dependencies through differentiated noise scales, while the inference step employs a novel sampling mechanism that dynamically adjusts these noise levels based on the current decoding state.

Experiments on synthetic data and structured reasoning benchmarks such as Countdown and Sudoku demonstrate that SPMDM substantially outperforms baseline MDMs and autoregressive models. Further evaluation on broader reasoning tasks, including commonsense reasoning and mathematical word problems, indicates competitive or superior performance compared to established diffusion and autoregressive models. The authors also perform ablation studies to highlight the importance of subsequence length and their novel sampling strategy.

**Questions:**

Here are my questions and suggestions for the authors to enhance the paper to help cross the acceptance barrier for me:

1. The authors introduce a subsequence-based noise scheduling approach, assigning distinct noise scales to fixed-length subsequences (as formulated in Equations 4–6). However, the paper primarily investigates uniform-length partitions. It would be valuable for the authors to explore and present explicit experimental comparisons with alternative partitioning schemes, such as semantically driven partitions  (based on syntactic structure, logical boundaries, or token-level uncertainty) or dynamically adjusted adaptive-length subsequences. Quantitative results comparing these alternative methods against the proposed uniform-length partitioning would substantially enhance the rigor and robustness of the presented approach.

2. The mathematical exposition around Equations (4–6) describing subsequence-specific noise scheduling currently lacks clarity in several important aspects. Could the authors explicitly clarify:
   - The exact mechanism for determining or optimizing the subsequence-dependent noise scale parameters ($t_k$) during training, and specifically how this formulation differs mathematically from standard masked diffusion schedules used in previous work (e.g., D3PM, MDLM)?
   - A detailed derivation or step-by-step intuition for these equations would greatly help me understand the precise modifications introduced by the authors, especially regarding how subsequence-level and token-level dependencies are mathematically decoupled or jointly modeled.

3. While the authors briefly mention computational overhead is minimal, the current paper does not provide a clear analysis or benchmark to support this claim. Could the authors present explicit computational complexity analyses or empirical runtime measurements demonstrating the overhead of managing multiple subsequence-specific noise schedules during training and inference, particularly compared to baseline masked diffusion models?

4. The authors achieve strong empirical results on structured reasoning tasks such as Countdown and Sudoku. However, it remains unclear to me whether these gains extend broadly to less structured, open-ended language generation tasks? Could the authors either provide additional experimental evaluations on such tasks (e.g., language modeling on standard benchmarks like WikiText or OpenWebText), or present a theoretical rationale explicitly discussing conditions under which the proposed subsequence partitioning approach is or is not expected to generalize effectively?

5. The paper mentions compatibility with adaptive sampling (Section 4.4) and briefly shows empirical improvements (Figure 3b). Could the authors expand on the conditions or theoretical intuition under which adaptive sampling interacts effectively with their subsequence-specific noise schedules?

**Ethical Concerns:**

["NO or VERY MINOR ethics concerns only"]

**Final Justification:**

I am happy to increase my score to a 5 as I believe the paper should be accepted. This should hopefully guarantee the acceptance of the paper.

**Limitations:**

The authors haven't really touched on potential negative societal impacts or broader limitations beyond technical issues. While the method itself seems pretty benign, it'd be helpful for me if they briefly addressed whether better logical reasoning abilities might accidentally make models more convincing at generating misinformation or biased outputs. A quick discussion acknowledging these risks and mentioning potential ways to mitigate them or responsibly deploy the model would strengthen the paper.

**Paper Formatting Concerns:**

None.

**Quality:**

3

**Strengths And Weaknesses:**

## Strengths
- The authors present an intuitive and nicely motivated improvement over standard masked diffusion approaches. They argue clearly (especially through Figure 1) that the ordering of token unmasking significantly affects generation quality, motivating their method of using subsequences with independently controlled noise levels.

- Overall, I find that the central methodological contribution (partitioning sequences into fixed-length subsequences and assigning subsequence-specific noise scales) is a fun, clever. and somewhat original take on masked diffusion models. Mathematically, this translates into adjusting the forward and backward processes (Eqns. 4–6) to explicitly model intra- and inter-subsequence dependencies, potentially addressing well-known shortcomings in traditional masked diffusion.

- Empirically, the results on challenging reasoning benchmarks like Countdown and Sudoku are quite compelling, significantly outperforming standard masked diffusion models (like MDLM, D3PM, SEDD), and even very competitive autoregressive baselines (GPT-2, LLaMA). Achieving near-perfect accuracy on Sudoku, for instance, really convinces me model’s ability to capture structural constraints effectively.

- Early motivation experiments (particularly Figure 1) are quite illustrative to me, clearly highlighting the advantages of carefully chosen sampling trajectories. These provide solid intuition behind the proposed changes.

## Weaknesses

-Despite a promising core idea, the paper is somewhat thin on (what could be) full methodological rigor. Compared to recent related works such as the cited "P2" paper (which conducts extensive, detailed ablations on various modeling choices) the current submission offers minimal systematic exploration. Beyond varying subsequence lengths (Figure 3a), the paper doesn’t investigate different noise schedules or alternative subsequence selection methods in sufficient depth. Given that their core innovation lies in subsequence-based partitioning and noise management, this kind of detailed analysis feels especially necessary.

- The core mathematical description could definitely use more detail and clearer exposition. While the math isn't incorrect (at least what I could define), these equations are currently presented in a somewhat terse manner. It would be helpful to step through at least one equation derivation or clearly explain each variable’s intuitive meaning, to aid readers less familiar with discrete diffusion modeling.

- While the results on structured tasks are quite nice, I'd appreciate if the authors would do more to clarify scalability and applicability beyond these specialized tasks. Specifically, it’s unclear if the subsequence-partitioning strategy will generalize equally well to less structured, more open-ended language generation problems.

- The authors briefly mention that the computational overhead is minimal, but this claim isn’t supported by quantitative details or analysis. Given the complexity of handling multiple noise schedules, some careful characterization of runtime, memory overhead, and scalability should be presented.

Overall, the paper presents an intriguing idea with strong empirical results on structured reasoning tasks but falls short of the rigorous experimentation and detailed mathematical clarity typical of top-tier diffusion modeling papers at NeurIPS.

---

> ### Author Rebuttal · Authors · 2025-07-31
>
> ### Q1. Other Partitioning Schemes
>
> Thank you for suggesting alternative segmentation strategies. This is indeed a promising direction. Indeed, we have explored similar approaches, including using punctuation marks as delimiters for text segmentation, which is similar to your proposed syntactic-structure-based partitioning.
> However, the main challenge is that such  syntactic structures are not available during inference with fully masked sequences.
>
> ### Q2. Formulation
>
> Thank you for your thoughtful consideration. We'd like to provide a comprehensive response to the concerns you raised.
>
> **About the subsequence-dependent noise scheduler.**
> For each subsequence $\mathbf{x}^k$, we independently sample noise scale parameters $t_k \sim \mathcal{U}[0, 1]$.
>
> The fundamental difference from standard masked diffusion models lies in the forward process formulation. In standard MDMs (D3PM, MDLM), the forward process is defined as:
> $$
> q_{t|0}(\mathbf{x}\_t^\ell \mid \mathbf{x}\_0^\ell) = \operatorname{Cat}(\alpha_t \mathbf{x}\_0^\ell + (1 - \alpha_t) \mathbf{m}).
> $$
>  In contrast, SPMDM's forward process operates at the subsequence level:
> $$
>   q_{\mathbf{t}\_k|0}(\mathbf{x}\_{t_k}^k \mid \mathbf{x}\_0^k) =  \prod_{\ell}^L q_{t_k|0}(\mathbf{x}\_{t_k}^{k,\ell} | \mathbf{x}\_{0}^{k,\ell}), \
>       q_{t_k|0}(\mathbf{x}\_{t_k}^{k,\ell} \mid \mathbf{x}\_0^{k,\ell}) = \operatorname{Cat}(\alpha_{t_k} \mathbf{x}\_0^{k,\ell} + (1 - \alpha_{t_k}) \mathbf{m}).
> $$
> This formulation introduces heterogeneous noise levels across subsequences within a single training sample.Meanwhile, the changes in the forward process also lead to modifications in the reverse process and the training objective (NELBO). We provide the detailed derivation and proof in Section 3.2 and Appendix B for your reference.
>
> **About the detailed derivation .** For Equation (4), the original process is $q_{t|0}(\mathbf{x}\_t^\ell \mid \mathbf{x}\_0^\ell) = \operatorname{Cat}(\alpha_t \mathbf{x}\_0^\ell + (1 - \alpha_t) \mathbf{m}),$
> and our modification is introduces heterogeneous noise levels across subsequences, which leads to
>  $q_{t_k|0}(\mathbf{x}\_{t_k}^{k,\ell} \mid \mathbf{x}\_0^{k,\ell}) = \operatorname{Cat}(\alpha_{t_k} \mathbf{x}\_0^{k,\ell} + (1 - \alpha_{t_k}) \mathbf{m}).$
>
> Following previous work, we derive the reverse process Equation (5) corresponding to the forward process by applying Bayes’ theorem.
>
> The modification above results in a different training objective (NELBO), as shown in Equation (6).
> A detailed derivation is provided in Appendix B.
> Since the derivation is somewhat complex, we may not have covered all the details. Please let us know if any parts are unclear.
>
> Here we provide a brief explanation of how our model captures inter-subsequence relationships. As reflected in our training loss:
> $$
>   \mathcal{L}\_{\text{NELBO}} = \sum_{k=1}^{K} \mathbb{E}\_{t_k \sim [0,1]} \mathbb{E}\_{q} \left[ \frac{\alpha_{t_k}'}{1 - \alpha_{t_k}} \log p_\theta(\mathbf{x}^k \mid \mathbf{x}\_{t_k}^k, \mathbf{x}\_\mathbf{t}^{-k}) \right],
> $$
> We shift the modeling unit from individual tokens to subsequences, and assign different noise scales to each subsequence. This design encourages the model to utilize information from other subsequences with lower noise levels, thereby facilitating the learning of inter-subsequence dependencies.
>
> ### Q3. Computational Overhead
>
> Thank you for raising the question regarding computational overhead. As discussed in Section 3.3, the potential computational overhead of our method compared to MDLM arises from two sources: (1) managing multiple subsequence-specific noise schedules (i.e. step 3 in Algo.1, step 5 in Algo.2), and (2) reintroducing the timestep embedding module (i.e. step 6 in Algo.2). Please kindly refer to the updated version of Algo.2 in our response to reviewer kRCe for clarification.
>
> Regarding the first point, managing separate noise schedules requires only basic matrix operations in PyTorch, which execute in constant time. Consequently, this overhead is negligible in practice.
>
> For the second point, we found that the official MDLM implementation does not actually remove the timestep embedder—it simply feeds it an all-zero input. Therefore, our reintroduction of timestep embedding incurs no additional computational cost compared to MDLM.
>
> To empirically validate SPMDM's efficiency, we conducted speed benchmarks on the Countdown dataset using 12K examples (batch size 32) on a single RTX 4090:
>
> |                      | MDLM  | SPMDM |
> | -------------------- | ----- | ----- |
> | Train (token/ms)     | 93.7  | 93.1  |
> | Eval (token/ms)      | 266.8 | 265.1 |
>
> For sampling (i.e., Algo.2), we measured the generation time for a single sample with 32 sampling steps. The results are reported in the table below:
>
> |                      | MDLM  | SPMDM |
> | -------------------- | ----- | ----- |
> | Sample (s)           | 0.55  | 0.57  |
>
> These results confirm that our method maintains computational efficiency comparable to baseline approaches while achieving superior performance.
>
> ### Q4. Open-ended Language Modeling
>
> As suggested by the reviewer, we now add an additional experiment on open-ended language modeling using the Text8 dataset (a small, character-level language modeling task).
>
> We follow [1] for network hyperparameters and dataset splits and compare with methods that employ a similar model size.
> Given the time constraints of the rebuttal period, we were limited to evaluating on this smaller dataset.
> The results are presented below (baseline performances are from [1]):
>
> | Method | BPC  |
> | ------ | ---- |
> | D3PM   | 1.45 |
> | SEDD   | 1.39 |
> | MDLM   | 1.38 |
> | SPMDM   | 1.37 |
>
> The results show that SPMDM achieves performance improvements on plain text generation, though the gains are relatively small. We attribute this to the nature of the Text8 dataset. Unlike reasoning tasks or structured datasets, Text8 lacks explicit inter-subsequence relationships that SPMDM is designed to exploit.
>
> Furthermore, uniform segmentation may inadvertently disrupt natural linguistic dependencies in Text8. This observation reinforces the importance of semantic or syntactic-based segmentation strategies mentioned in your earlier comment (Q1).
>
> ### Q5. Effectiveness of Adaptive Sampling
>
> Thank you for your thoughtful consideration.
> As discussed in Section 3.1, the subsequence-specific noise schedule in SPMDM is designed to guide the sampling trajectory to better align with the simple path characteristics.
> Similarly, adaptive sampling (including both top-K probability sampling and top-K probability margin sampling) aims to unmask the easier and/or more confident tokens first, and thus also encourages the model to follow simpler paths as demonstrated in Figure 1.
> In other words, both the subsequence-specific noise schedules and the adaptive sampling strategy are motivated by the same underlying principle. Therefore, combining the two can enhance the model's overall performance, as shown in figure 3(b).
>
>
> ### References
>
> [1] Simple Guidance Mechanisms for Discrete Diffusion Models

---

> > ### Comment · Reviewer_E7mR · 2025-08-01
> > **Thank you!**
> >
> > The authors do a strong job of responding to my comments, and I am happy to hold my score at a 4, which reflects my desire for the paper to be accepted, but also the novelty of the methodology. I would encourage the AC to take this into account.

---

> > > ### Comment · Reviewer_E7mR · 2025-08-05
> > >
> > > I would like to note that I increased my score to a 5, not just maintained it as a 4. I hope the AC will consider accepting the paper.

---

### Official Review · Reviewer_5B9g · 2025-06-28

**Clarity:** 3
**Significance:** 3
**Originality:** 3
**Rating:** 4
**Confidence:** 3

**Summary:**

This paper introduces the Simple Path Mask Diffusion Model (SPMDM), improving masked diffusion models performance by simplifying sampling paths. The authors identify two key characteristics of efficient sampling: prioritizing tokens near unmasked ones and generating earlier reasoning subsequences. Comprehensive experiments show that SPMDM outperforms existing diffusion and autoregressive models, effectively capturing structural rules and enabling parallel and controllable generation.

**Questions:**

1. How was the optimal subsequence length (e.g., L=9 for Sudoku) determined? Why not use some regular values, e.g. 16, 32, but using like 9, 81 and 162 for Sudoku?

2. How do you consider applying the proposed model to variable-length generation scenarios? For example, Block Diffusion appends blocks of tokens to the end of the generated sequence after the entire sequence is unmasked, while in the parallel sampling proposed in the paper, parallel sampling requires multiple blocks that are not completely unmasked, which is kind of contradictory.

**Ethical Concerns:**

["NO or VERY MINOR ethics concerns only"]

**Final Justification:**

I am satisfied with the rebuttal.

**Limitations:**

Yes

**Paper Formatting Concerns:**

Nil

**Quality:**

3

**Strengths And Weaknesses:**

Strengths
1. SPMDM partitions sequences into subsequences with distinct noise scales, encouraging the model to learn intra-subsequence token dependencies and inter-subsequence logical relationships. This design addresses MDMs’ training challenges where denoising difficulty varies with unmasked token positions .
2. The paper identifies critical sampling path characteristics (local neighborhood prioritization and logical-order generation), which align with human problem-solving strategies. These insights guide the proposed model’s training and sampling to facilitate more efficient unmasking orders.
3. SPMDM achieves state-of-the-art performance on complex tasks, outperforming auto-regressive and diffusion baselines. It handles large combinatorial spaces and long-range reasoning better due to its subsequence-based dependency modeling.

Weaknesses
1. When subsequence length is too short (e.g., L=4), performance drops because subsequences lack sufficient semantic or logical information, hindering inter-subsequence dependency modeling. In addition, long subsequences (e.g., L=16) reduce the need for cross-subsequence reasoning.
2. There is a lack of a specific method to set up the subsequence length.

---

> ### Author Rebuttal · Authors · 2025-07-31
>
> ### W1, W2, Q1. Optimal Subsequence Length
>
> Thank you for your insightful comments.
>
> For datasets with clear structural patterns (e.g., Sudoku), we leveraged the inherent structure for segmentation. Specifically, since Sudoku inputs are 9×9 grids, we used the row length (i.e., 9) as the basis for subsequence segmentation. Here, the question length is 81 (9×9), and we segment the concatenated sequence of question and solution using a subsequence length of 81. As noted in Section 4.4, when the subsequence length is set to 162 (i.e., 81 + 81), the model effectively reduces to MDLM.
>
> For datasets without obvious structure (e.g., Countdown, GSM8K), we used a fixed subsequence length of 8. Tables 2 and 4 demonstrate that this simple choice yields strong performance across all datasets.
>
> We acknowledge that optimal sequence length selection is important. We are currently investigating adaptive methods and will include results in the final version if they improve performance.
>
> ### Q2. Variable-length Generation Scenarios
>
> Thank you for this insightful question. You are correct that our proposed SPMDM method cannot be directly applied to variable-length generation scenarios like Block Diffusion [1].
>
> The key difference is that Block Diffusion handles variable-length generation through autoregressive modeling of inter-block dependencies, which SPMDM currently lacks. However, as shown in Tables 1-3, our method achieves superior performance compared to Block Diffusion due to its full bidirectional modeling capability, while Block Diffusion is limited by its semi-unidirectional approach and incurs higher computational costs during training.
>
> We acknowledge that variable-length generation is an important problem. We are actively exploring approaches to extend our method to generate high-quality variable-length sequences while maintaining the performance advantages of SPMDM.
>
> ### References
>
> [1] Block Diffusion: Interpolating Between Autoregressive and Diffusion Language Models

---

> > ### Comment · Reviewer_5B9g · 2025-08-07
> >
> > Thanks for your response. I remain positive with the paper.

---

### Official Review · Reviewer_kRCe · 2025-07-01

**Clarity:** 3
**Significance:** 2
**Originality:** 2
**Rating:** 4
**Confidence:** 3

**Summary:**

This paper highlights the importance of sampling paths in masked diffusion models, emphasizing that effective generation should prioritize tokens near those already decoded and substrings that appear earlier in the reasoning process. To enforce these principles, the authors propose the Simple Path Mask Diffusion Model (SPMDM), which partitions the input sequence into subsequences and assigns each a distinct noise scale. Experiments on toy examples, problem-solving tasks, and a range of reasoning datasets demonstrate that SPMDM achieves superior performance.

**Questions:**

Can the proposed sampling method be directly applied to a pretrained MDLM? It appears that it can, and it would be valuable to understand how well the method performs in a training-free setting.

**Ethical Concerns:**

["NO or VERY MINOR ethics concerns only"]

**Final Justification:**

My initial concern was that the baseline did not include sampling methods commonly adopted in the field. Since this issue has been resolved in the rebuttal, I am raising my score from 3 to 4. However, as I do not find the proposed approach to be particularly innovative or impactful, I am not assigning a higher score.

**Limitations:**

yes

**Quality:**

2

**Strengths And Weaknesses:**

## Strengths
1. The sampling path is a critical research topic for masked diffusion models (MDMs).

2. The paper is clearly written and logically structured, making it easy to follow.

3. The proposed method is well-motivated, intuitive, and straightforward to implement.

## Weakness
The main limitation of this paper lies in its choice of baselines. Sampling strategies such as MaskGIT [1] (also referred to as Top-K probability sampling in [2]) and Top-K probability margin sampling [2] are widely adopted in MDM research. Importantly, these strategies can be applied to pretrained MDMs in a training-free manner. Given their relevance and practicality, they should serve as direct baselines for this work. However, the authors did not include comparisons with these methods.

[1] Chang et al. MaskGIT: Masked Generative Image Transformer.

[2] Kim et al. Train for the Worst, Plan for the Best: Understanding Token Ordering in Masked Diffusions.

---

> ### Author Rebuttal · Authors · 2025-07-31
>
> ### W1. Choice of Baselines
>
> Thank you for your valuable feedback. We apologize for the misunderstanding and recognize that our sampling procedure (Algorithm 2) may not have been clearly written in the manuscript.
> The following shows the revised Algorithm 2. In particular, we change $p_\theta(\cdot \mid \mathbf{x}\_{t_k}^{k}, \mathbf{x}\_{\mathbf{t}}^{-k})$ to $p_\theta(\cdot \mid \mathbf{x}\_{t_k}^{k}, \mathbf{x}\_{\mathbf{t}}^{-k},t_k)$ so as to explicitly show its dependence on the time step $t_k$.
>
> ***
> $$
> \begin{aligned}
> 1.\ & \textbf{Input:} \text{Network}\ p_\theta, \text{subsequence length}\ L, \text{time}\ [0, 1], \text{sampling steps}\ T, \text{ordering oracle function}\ \mathcal{F} \\\\
> 2.\ & \textbf{Initialize}\ \mathbf{x}\_\mathbf{1} \sim \{\mathbf{m}\}^N,\ \mathbf{t} \gets \mathbf{1},\ \Delta t \gets 1 / T \\\\
> 3.\ & \textbf{for } n=0 \text{ to } T \textbf{ do} \\\\
> 4.\ & \quad \forall k: \text{Count unmasked tokens } n_k \\\\
> 5.\ & \quad \forall k: t_k \gets 1 - n_k /L \\\\
> 6.\ & \quad \forall k: \hat{\mathbf{x}}\_{0}^{k} \sim p_\theta(\cdot \mid \mathbf{x}\_{t_k}^{k}, \mathbf{x}\_{\mathbf{t}}^{-k}, t_k) \\\\
> 7.\ & \quad \textbf{if} \text{ using adaptive sampling strategy } \textbf{then} \\\\
> 8.\ & \quad \quad \text{Sample a set of masked token indices } \mathcal{S} = \mathcal{F}(\theta, \mathbf{x}\_\mathbf{t}) \\\\
> 9.\ & \quad \quad \forall (i, \ell) \in \mathcal{S}: \mathbf{x}\_\mathbf{t}^{i, \ell} = \hat{\mathbf{x}}\_\mathbf{0}^{i, \ell} \\\\
> 10.\ & \quad \textbf{else} \\\\
> 11.\ & \quad \quad \forall k: s_k \gets \max(t_k - \Delta t, 0) \\\\
> 12.\ & \quad \quad \forall k: \text{For all masked tokens, with probability } \frac{s_k}{t_k}, \hat{\mathbf{x}}\_\mathbf{0}^{k, \ell} \gets \mathbf{m} \\\\
> 13.\ & \quad \quad \text{Update } \mathbf{x}\_\mathbf{t} \gets \hat{\mathbf{x}}\_\mathbf{0} \\\\
> 14.\ & \quad \textbf{end if} \\\\
> 15.\ & \textbf{end for} \\\\
> 16.\ & \textbf{Return } \mathbf{x}\_\mathbf{t}
> \end{aligned}
> $$
> ***
>
> Moreover, in the original submission, experimental results on the proposed method (denoted "Ours" in section 4.4) use steps 11-13 (note that these steps are directly inherited from the MDLM [1]), while the experimental results on the method denoted "Ours + Adaptive" in Figure 3(b) use steps 7-9 with top-K probability sampling.
> In other words, the commonly-used adaptive sampling strategy mentioned by the reviewer has already been used as a baseline.
>
> As suggested by the reviewer, we now conduct additional experiment with top-K probability margin sampling, using the setup as in Section 4.4 (Figure 3(b)).
> For easier comparison, we convert Figure 3(b) to the table below, and add in the results on top-K probability margin sampling (note that 'Ours + Adaptive' in figure 3(b) now corresponds to the entry 'Ours + Top-K Probability Sampling').
>
> | Task      | Sampling Strategy                        | Accuracy (%) |
> | --------- | ---------------------------------------- | ------------ |
> | CountDown | DDPM                                     | 87.9         |
> | CountDown | Ours                                     | 90.2         |
> | CountDown | Ours + Top-K probability sampling        | 92.7         |
> | CountDown | Ours + Top-K probability margin sampling | 92.9         |
> | Sudoku    | DDPM                                     | 95.4         |
> | Sudoku    | Ours                                     | 99.9         |
> | Sudoku    | Ours + Top-K probability sampling        | 100.0        |
> | Sudoku    | Ours + Top-K probability margin sampling | 100.0        |
>
> As can be seen, while 'Ours + Top-K probability margin sampling' achieves the best performance, the improvement over 'Ours + Top-K probability sampling' is small. We will add this new comparison and further clarifications in the final version.
>
> ### Q1. Training-free Setting Performance
>
> Thank you for your question. As suggested, we also add an experiment on the training-free setting with a pretrained MDLM.
> However, as mentioned in the response above,
> $p_\theta(\cdot \mid \mathbf{x}\_{t_k}^{k}, \mathbf{x}\_{\mathbf{t}}^{-k},t_k)$
> depends on the time step $t_k$.
> However, in the standard MDLM, time step $t_k$ is NOT used in the training. Hence, in order to be used
> on a pretrained MDLM, we do not feed the time step into $p_\theta$ (of the pretrained MDLM).
> The algorithm (modified from the algorithm above) is shown in the following:
>
> ***
> $$
> \begin{aligned}
> 1.\ & \textbf{Input: } \text{Network}\ p_\theta, \text{subsequence length}\ L, \text{time}\ [0, 1], \text{sampling steps}\ T \\\\
> 2.\ & \textbf{Initialize}\ \mathbf{x}\_\mathbf{1} \sim \{\mathbf{m}\}^N,\ \mathbf{t} \gets \mathbf{1},\ \Delta t \gets 1 / T \\\\
> 3.\ & \textbf{for } n=0 \text{ to } T \textbf{ do} \\\\
> 4.\ & \quad \hat{\mathbf{x}}\_{\mathbf{0}} \sim p_\theta(\cdot \mid \mathbf{x}\_{\mathbf{t}}) \\\\
> 5.\ & \quad \forall k: \text{Count unmasked tokens } n_k \\\\
> 6.\ & \quad \forall k: t_k \gets 1 - n_k /L \\\\
> 7.\ & \quad \forall k: s_k \gets \max(t_k - \Delta t, 0) \\\\
> 8.\ & \quad \forall k: \text{For all masked tokens, with probability } \frac{s_k}{t_k}, \hat{\mathbf{x}}\_\mathbf{0}^{k, \ell} \gets \mathbf{m} \\\\
> 9.\ & \quad \text{Update } \mathbf{x}\_\mathbf{t} \gets \hat{\mathbf{x}}\_\mathbf{0} \\\\
> 10.\ & \textbf{end for} \\\\
> 11.\ & \textbf{Return } \mathbf{x}\_\mathbf{t}
> \end{aligned}
> $$
> ***
>
> The following table shows the testing on the Countdown dataset.
> For comparison, we also include the proposed SPMDM below.
>
> |               | Models | Sampling Strategy                 | Accuracy (%) |
> | ------------- | ------ | --------------------------------- | ------------ |
> | Training-free | MDLM   | DDPM                              | 75.5         |
> | Training-free | MDLM   | Top-K probability sampling        | 85.8         |
> | Training-free | MDLM   | Top-K probability margin sampling | 89.4         |
> | Training-free | MDLM   | Ours                              | 79.2         |
> | With Training | SPMDM  | Ours                              | 90.2         |
>
> As can be seen, while the proposed subsequence-based sampling method can be applied on a pretrained MDLM and yields performance improvements, the gains are significantly smaller than those obtained by the proposed SPMDM (which modifies both both the training and sampling phases).
> The results and discussion here will be included in the final version.
>
>
> ### References:
>
> [1] Simple and Effective Masked Diffusion Language Models

---

> > ### Comment · Reviewer_kRCe · 2025-08-04
> >
> > Thank you for the response. I believe that this new comparison in the rebuttal undoubtedly improves the clarity of the paper. I decide to increase my score to 4.

---

### Official Review · Reviewer_kCs5 · 2025-07-03

**Clarity:** 3
**Significance:** 4
**Originality:** 3
**Rating:** 5
**Confidence:** 4

**Summary:**

The paper proposes a novel framework called Simple Path Mask Diffusion Model (SPMDM) to address challenges in masked diffusion models (MDMs) for discrete sequence generation. While MDMs enable flexible, parallel, and any-order generation, their training is more complex than autoregressive models (AR) due to variations in token prediction difficulty based on the positions of unmasked tokens.

Through analysis of adapative sampling on mutliple tasks. The authors identify two key patterns of efficient sampling paths: prioritizing tokens near unmasked ones and generating subsequences earlier in reasoning.

Based on this observation, the authors propose SPMDM, which improves upon existing MDMs by partitioning input sequences into fixed-length, non-overlapping subsequences and applying varying noise scales. This approach encourages better token-level and cross-subsequence dependency modeling.

**Questions:**

see weakness

**Ethical Concerns:**

["NO or VERY MINOR ethics concerns only"]

**Limitations:**

yes

**Paper Formatting Concerns:**

The paper is well-formatted.

**Quality:**

4

**Strengths And Weaknesses:**

Strengths:

1. Novel analysis on sampling of MDM: The paper provides a novel analysis of sampling paths in MDMs, identifying two key characteristics that define simpler and more efficient sampling strategies. This analysis provides a nice insight of how sampling paths influence both training efficiency and inference quality in MDMs.

2. The model performs well across diverse tasks, including both general NLP tasks as well as tasks requiring combinatorial reasoning (Countdown) and structural consistency (Sudoku), showcasing its versatility in handling complex reasoning and planning problems.

3. The architecture is designed to scale efficiently, showing strong performance even with smaller parameter budgets (e.g., 6M and 85M models), which is beneficial for resource-constrained environments.

Weakness:

1. How does SPMDM perform in comparison with block diffusion methods [1]? Block diffusion methods also break generation into chunks and model the relationship both between and within the chunks.

2. The motivation emphasizes learning tokens in the neighborhood, which is improved by using subsequence-level diffusion. For planning tasks, the priority is to learn earlier reasoning tokens. Does this learning order also appear in SPMDM's inter-subsequence relationships? Additionally, SPMDM supports any-order subsequence relationships. How does the preferred order differ for different tasks, e.g., Countdown tasks vs. commonsense reasoning?

[1] Arriola, Marianne, et al. "Block diffusion: Interpolating between autoregressive and diffusion language models." ICLR 2025.

---

> ### Author Rebuttal · Authors · 2025-07-31
>
> ### Q1. How does SPMDM perform in comparison with block diffusion methods?
>
> Thank you for your thoughtful comments. The key difference is:
> Block Diffusion (BDLM) [1] methods use autoregressive modeling, while SPMDM employs different noise scheduling across subsequences. BDLM is semi-unidirectional rather than fully bidirectional. While this may benefit general text modeling, it negatively impacts reasoning tasks that require bidirectional context understanding.
>
>  Additionally, BDLM's hybrid diffusion-AR approach incurs higher training costs, whereas SPMDM adds minimal training overhead while achieving superior performance on reasoning tasks.
>
> As for the empirical performance, the following is a summary of the observations from the real-world tasks.
> On Countdown tasks (Table 2), SPMDM achieves 0.9%, 7.2%, and 7.9% higher accuracy than BDLM on CD-3, CD-4, and CD-5 respectively. On Sudoku (Table 3), SPMDM surpasses BDLM by 8.1% accuracy.
>
> We will add the above discussion and more detailed comparison with BDLM in the final version.
>
> ### Q2. For planning tasks, the priority is to learn earlier reasoning tokens. Does this learning order also appear in SPMDM's inter-subsequence relationships?
>
> Thank you for your question. Yes, this learning order do appear in SPMDM's inter-subsequence relationships. To demonstrate
> how SPMDM captures inter-subsequence relationships through its denoising order, we now add the following experiment on the Countdown task.
>
> First, we pick an example from the Countdown-3 test sets and show the step-by-step denoising process of SPMDM as follows:
>
> ```
> 1,82,8 ### 73 ### [MASK]2-[MASK][MASK][MASK][MASK][MASK][MASK]1[MASK][MASK]=[MASK][MASK]
> 1,82,8 ### 73 ### [MASK]2-1[MASK][MASK][MASK][MASK][MASK]1[MASK][MASK]=[MASK][MASK]
> 1,82,8 ### 73 ### 82-1[MASK]8[MASK][MASK][MASK]1[MASK]8=[MASK][MASK]
> 1,82,8 ### 73 ### 82-1[MASK]8[MASK][MASK][MASK]1[MASK]8=7[MASK]
> 1,82,8 ### 73 ### 82-1[MASK]81[MASK][MASK]1[MASK]8=7[MASK]
> 1,82,8 ### 73 ### 82-1=81[MASK][MASK]1[MASK]8=7[MASK]
> 1,82,8 ### 73 ### 82-1=81,[MASK]1-8=7[MASK]
> 1,82,8 ### 73 ### 82-1=81,[MASK]1-8=73
> 1,82,8 ### 73 ### 82-1=81,81-8=73
> ```
>
> As can be seen, SPMDM first decodes the first question, followed by the second equation.
>
> In addition, we conduct a quantitative experiment on the Countdown test sets.
> Specifically, following the experimental setup in Section 4.2, we set the sequence length to 48 and the subsequence length to 8, using the CD-3 problem as the prompt input. We calculate the average denoising timestep for tokens within each subsequence throughout the generation process.
>
> We set the number of sampling steps to 12. The experimental results are presented in the table below, where *Index* indicates the subsequence index, and *Avg Unmask Steps* refers to the average denoising timestep of tokens within each subsequence:
>
> | Index | Avg. Unmask Timesteps |
> | ----- | --------------------- |
> | 0     | 0.0                   |
> | 1     | 0.0                   |
> | 2     | 4.58                  |
> | 3     | 5.29                  |
> | 4     | 6.37                  |
> | 5     | 6.25                  |
>
> Note that, based on the current partition with a subsequence length of 8, the first two subsequences are prompts. As a result, for sequence index 0 and 1, the average unmask steps are always zero.
>
> We observe that SPMDM prioritizes denoising the first equation (i.e, sequence index 2).
> This behavior is consistent with construction process of Countdown ground-truth solutions.
> The process begins by solving the first equation, which helps determine the valid factors that can be used in subsequent equations. The last two subsequences mainly consist of padding tokens, and thus do not exhibit any significant differences in denoising order.
>
> Therefore, we believe that SPMDM successfully captures the underlying dependencies between subsequences.
>
> ### Q3. How does the preferred order differ for different tasks?
>
> To illustrate this, we perform the experiment in Q2 above on the Sudoku task.
> Recall that  each row in the Sudoku question is treated as a subsequence.
> We set the number of sampling steps to 18, and the results are shown in the table below.
>
> | Index | Avg. Unmask Timesteps |
> | ----- | --------------------- |
> | 0     | 9.43                  |
> | 1     | 9.46                  |
> | 2     | 9.51                  |
> | 3     | 9.74                  |
> | 4     | 9.72                  |
> | 5     | 9.83                  |
> | 6     | 9.70                  |
> | 7     | 9.67                  |
> | 8     | 9.42                  |
>
> As can be seen, the average denoising timestep of the subsequences exhibits a uniform distribution, which is different from that on the Countdown tasks we saw above.
> This is because the rows in the Sudoku question vary in difficulty, thus prioritizing the denoising of simpler rows (subsequences) leads to a simpler overall sampling path.
> Given a sufficiently large amount of data, the difficulty distribution across rows is likely to be approximately uniform.
>
> Overall, combining with the observations from Q2 above, we find that  SPMDM  prefers prioritizing the denoising of subsequences that are easier to complete.
>
>
>
> ### References
>
> [1] Block Diffusion: Interpolating Between Autoregressive and Diffusion Language Models

---

> > ### Comment · Reviewer_kCs5 · 2025-08-01
> > **Thank you for your response**
> >
> > The added analysis are all very interesting. I keep my rate of accepting this work.

---

### Decision · Program_Chairs · 2025-09-17

**Decision:**

Accept (poster)

**Comment:**

The paper proposes new improved to masked diffusion models (MDMs), which is an interesting alternative to the autoregressive model for language generation and reasoning tasks. The improvement is on proposing a heuristic that improve the decoding order of the MDMs. The experimental results demonstrate the effectiveness of the proposed method. The reviewers all acknowledge the contribution of the paper, and after an effective rebuttal and discussion period, the reviewers are satisfied with the authors' feedback.

Therefore, I recommend the paper for acceptance at NeurIPS.

AC.